# Assessing Quality of Life in Hemodialysis Patients in Kazakhstan: A Cross-Sectional Study

**DOI:** 10.3390/jcm14145021

**Published:** 2025-07-16

**Authors:** Aruzhan Asanova, Aidos Bolatov, Deniza Suleimenova, Yelnur Khazhgaliyeva, Saule Shaisultanova, Sholpan Altynova, Yuriy Pya

**Affiliations:** 1Department of Science, “University Medical Center” Corporate Fund, Astana 010000, Kazakhstan; asanova.aruzhan@umc.org.kz (A.A.); yelnur.khazhgaliyeva@nu.edu.kz (Y.K.); 2School of Medicine, Shenzhen University, Shenzhen 518060, China; 3Department of Child Nephrology, Dialysis and Transplantation, Clinical Academic Department of Pediatrics, “University Medical Center” Corporate Fund, Astana 010000, Kazakhstan; deniza.suleimenova@nu.edu.kz; 4School of Medicine, Nazarbayev University, Astana 010000, Kazakhstan; 5Department of Medical and Regulatory Affairs, “University Medical Center” Corporate Fund, Astana 010000, Kazakhstan; shaisultanova.s@umc.org.kz (S.S.); venera.altynova@umc.org.kz (S.A.); 6Clinical Academic Department of Cardiac Surgery, “University Medical Center” Corporate Fund, Astana 010000, Kazakhstan; yuriy.pya@umc.org.kz

**Keywords:** hemodialysis, quality of life, validation, predictors, Kazakhstan

## Abstract

**Background**: The Kidney Disease and Quality of Life Short Form (KDQOL-SF™ 1.3) is widely used to assess health-related quality of life (HRQoL) in patients with end-stage renal disease. However, no prior validation had been conducted in Kazakhstan, where both Kazakh and Russian are commonly spoken. This study aimed to validate the Kazakh and Russian versions of the KDQOL-SF™ 1.3 and to identify predictors of HRQoL among hemodialysis patients in Kazakhstan. **Methods**: A cross-sectional survey was conducted among 217 adult hemodialysis patients from February to April 2025 using a mixed-methods approach (in-person interviews and online data collection). Psychometric testing included Cronbach’s alpha, floor and ceiling effect analysis, and Pearson correlations with self-rated overall health. Multiple linear regression was used to identify predictors of the Kidney Disease Component Summary (KDCS), Physical Component Summary (PCS), and Mental Component Summary (MCS) scores. **Results**: Both language versions demonstrated acceptable to excellent internal consistency (Cronbach’s α = 0.692–0.939). Most subscales were significantly correlated with self-rated health, supporting construct validity. Regression analyses revealed that greater satisfaction with care, better economic well-being, and more positive dialysis experiences were significant predictors of higher KDCS and MCS scores. Lower PCS scores were associated with female gender, comorbidities, and financial burden. Importantly, financial hardship and access challenges emerged as strong negative influences on HRQoL, underscoring the role of socioeconomic and care-related factors in patient well-being. **Conclusions**: The KDQOL-SF™ 1.3 is a valid and reliable tool for assessing quality of life among Kazakh- and Russian-speaking hemodialysis patients in Kazakhstan. Integrating this instrument into routine clinical practice may facilitate more personalized, patient-centered care and help monitor outcomes beyond traditional clinical indicators. Addressing economic and access-related barriers has the potential to significantly improve both physical and mental health outcomes in this vulnerable population.

## 1. Introduction

Chronic kidney disease (CKD) is a global public health problem that affects approximately 13.4% of the world’s population. Every year, between 4.9 and 7.1 million individuals progress to end-stage renal disease (ESRD), a severe and life-threatening stage of CKD that requires intensive medical intervention [1]. Kidney transplantation is a treatment of choice for people with ESRD, providing substantial enhancements in quality of life (QoL). However, the availability of donor organs remains a critical barrier in many countries, leading to prolonged waiting times for transplantation. According to national statistics, approximately 11,000 patients receive dialysis in Kazakhstan, yet only 3748 are currently registered on the kidney transplant waiting list. This highlights the limited availability of transplantation as a treatment option and underscores the importance of optimizing quality of life for patients maintained on long-term dialysis. Consequently, many individuals with ESRD must rely on dialysis as a critical, life-sustaining therapy to replace lost kidney function.

In Kazakhstan, data from a nationwide registry covering the years 2014 to 2018 showed a substantial increase in the number of dialysis patients, rising from 135.2 to 350.2 per million population, with a notable majority receiving hemodialysis (98.7%) and a mortality rate that decreased significantly over the years [2]. Despite its life-sustaining benefits, dialysis profoundly impacts patients’ overall well-being due to its rigorous nature and associated health issues, often resulting in a diminished QoL characterized by unpleasant symptoms and complex treatment processes [3].

Dialysis significantly impacts not only the physical well-being of patients but also their emotional, mental, and social health. Physically, those undergoing dialysis often suffer from fatigue, anemia, and various issues arising from both ESRD and the dialysis treatment, all of which can lower their QoL. Mentally, the repetitive and demanding nature of dialysis is strongly associated with increased rates of psychological distress, including depression and anxiety. This is often exacerbated by the chronic nature of kidney disease, uncertainty about the future, and the lifestyle changes necessitated by ongoing treatment [4]. Furthermore, prolonged dialysis can negatively affect emotional well-being, including aspects like sexual satisfaction and cognitive abilities, particularly those related to organization and planning, which are essential for maintaining a sense of personal well-being. Socially, dialysis patients frequently report decreased levels of social interaction and support, often feeling isolated or burdensome to family members [5]. These multifaceted challenges highlight the urgent need for validated and culturally sensitive tools to assess health-related quality of life (HRQoL) in dialysis patients, particularly in under-researched regions such as Central Asia. Accurate assessment is crucial for informing healthcare policies and improving patient-centered care in these regions.

In recent years, the management of ESRD has undergone significant evolution, emphasizing not only survival but also the improvement of HRQoL. Beyond dialysis adequacy, relational skills and communication competencies of nephrology nurses are now recognized as critical determinants of patient outcomes, fostering trust, adherence, and emotional support [6,7]. Evidence suggests that interventions focusing on empathetic communication and shared decision-making can lead to tangible improvements in treatment adherence and reduce psychological distress among hemodialysis patients. In parallel, technological innovations, including wearable dialysis devices, home-based hemodialysis systems, and biofeedback-controlled ultrafiltration, aim to minimize symptom burden and enhance lifestyle flexibility, supporting better physical and mental well-being [8,9]. Moreover, international guidelines increasingly highlight the role of conservative kidney management and structured pre-transplant preparation, including patient education and early referral for transplantation, as integral parts of comprehensive ESRD care [10,11]. These advances underscore the shift toward patient-centered and holistic approaches that address the complex interplay of medical, psychosocial, and economic factors influencing HRQoL. Incorporating culturally adapted, validated tools such as the KDQOL-SF™ is therefore essential to capture these multidimensional outcomes and guide individualized care strategies in diverse settings like Kazakhstan.

The Kidney Disease Quality of Life Short Form (KDQOL-SF™ 1.3) is a comprehensive and widely recognized tool designed to assess the multidimensional impact of kidney disease on patients’ lives. It incorporates various subscales, including symptoms specific to the disease (e.g., fatigue, pain, and the burden of kidney disease) and domains of general well-being (e.g., physical function, emotional well-being, and social support) [12]. Globally, numerous studies utilizing the KDQOL-SF™ 1.3 have helped identify consistent areas requiring targeted intervention. While patients often report relatively high satisfaction with dialysis team support and levels of social interaction [13], they consistently rate emotional and physical health domains as the lowest. These findings underscore persistent gaps in the provision of mental health services and physical rehabilitation for dialysis patients.

Furthermore, QoL outcomes are significantly influenced by socio-demographic and clinical variables. Age, education level, and comorbidities such as diabetes or hypertension are known to contribute to disparities in patient-reported outcomes, with older, less-educated individuals and those with multiple comorbidities reporting poorer QoL [13,14]. In this context, linguistic and cultural adaptation of the KDQOL-SF™ 1.3 becomes crucial. Successful validations in countries such as Iran, Egypt, and the Philippines have demonstrated that culturally tailored versions of the instrument more accurately reflect local patient experiences [15,16]. In Kazakhstan, where both Kazakh and Russian are spoken, ensuring linguistic equivalence is essential for capturing the perspectives of all patient groups.

By addressing these gaps, this study provides the first evidence on the applicability of the KDQOL-SF™ in Kazakhstan and contributes to a better understanding of factors influencing patient-reported outcomes in a multilingual dialysis context. The aims of this study are (1) to validate the Kazakh and Russian versions of the KDQOL-SF™ 1.3 among adult patients receiving hemodialysis in Kazakhstan, and (2) to identify sociodemographic, clinical, and experiential predictors of HRQoL in this population.

## 2. Materials and Methods

### 2.1. Study Aims and Research Questions

The primary objective of this study was to validate the Kazakh and Russian versions of the KDQOL-SF™ 1.3 among adult patients receiving hemodialysis in Kazakhstan. The secondary objective was to identify sociodemographic, clinical, and experiential predictors of health-related quality of life in this population.

Based on these objectives, the specific research questions were:

(1) Are the Kazakh and Russian versions of the KDQOL-SF™ 1.3 reliable and valid for assessing HRQoL among hemodialysis patients in Kazakhstan?

(2) Which sociodemographic, clinical, and care-related factors are associated with variations in KDCS, PCS, and MCS scores among this population?

### 2.2. Study Design and Participants

This was a cross-sectional study conducted among adult patients receiving maintenance hemodialysis in multiple dialysis centers across Kazakhstan. A total of 217 participants were recruited between 24 February and 28 April 2025, using convenience sampling. Eligibility criteria included age ≥ 18 years, a confirmed diagnosis of end-stage renal disease (ESRD), current enrollment in hemodialysis for at least three months, and the ability to read and complete the questionnaire in either Kazakh or Russian.

Data collection was conducted using a mixed-methods approach. Patients were recruited and surveyed in two ways: (1) through in-person interviews conducted at dialysis centers in Astana and (2) via an online questionnaire, which was distributed by sharing a survey link in WhatsApp messenger patient support group chats. This strategy allowed broader geographic reach and improved participation among patients across urban and remote areas. To minimize potential bias, the same standardized questionnaire and instructions were used for both in-person and online surveys, and participants were assured of confidentiality and encouraged to respond candidly.

This study was conducted and reported in accordance with the Strengthening the Reporting of Observational Studies in Epidemiology (STROBE) guidelines [17]. The completed STROBE checklist is provided as Appendix A.

### 2.3. Research Instruments

Participants completed a structured questionnaire that included both sociodemographic characteristics and patient-reported measures related to dialysis experience, treatment adherence, and financial burden.

Sociodemographic variables included age, gender, ethnicity, language preference (Kazakh or Russian), place of residence, educational attainment, marital status, presence of children (yes/no), and employment status. Clinical variables included self-reported cause of end-stage renal disease, duration of dialysis treatment, and list of comorbidities.

The Kidney Disease and Quality of Life-Short Form (KDQOL-SF™, Santa Monica, CA, USA) version 1.3 was used to assess health-related quality of life [12]. This validated instrument includes a kidney disease-specific core and a generic core (SF-36), covering domains such as physical functioning, mental health, symptoms, burden of disease, and dialysis staff interaction. The scale was translated and culturally adapted into Kazakh using standard forward-backward translation procedures. The Russian version used was previously validated among patients in Russia by Dr. Vassilieva [18]. The KDQOL-SF™ yields three summary scores: Kidney Disease Component Summary (KDCS), Physical Component Summary (PCS), and Mental Component Summary (MCS). Additionally, a global health rating item was used for assessing perceived overall health.

In addition, several dialysis-related experience variables were collected.

Ease of access to the dialysis center was assessed with the question “How easy is it for you to get to the dialysis center?” Responses were rated on a 5-point scale: Very difficult, Somewhat difficult, Neither easy nor difficult, Somewhat easy, and Very easy.

Experience with dialysis treatment was measured with the item “How would you rate your experience with dialysis treatment?” Responses included Very positive, Positive, Neutral, Negative, and Very negative.

Missed dialysis sessions were assessed with the question “How often have you missed dialysis sessions in the past 3 months?” Response options were Never, 1–2 times, and 3 or more times.

Dietary adherence difficulty was assessed with the item “How often do you find it difficult to follow dietary restrictions?” Participants responded using a 5-point scale: Never, Rarely, Sometimes, Often, and Always.

Satisfaction with medical care was assessed with the question “How satisfied are you with the quality of medical care provided at the dialysis center?” Response options included Very satisfied, Satisfied, Somewhat satisfied, Dissatisfied, and Very dissatisfied.

Economic well-being was measured on a 5-point Likert-type scale, ranging from 1 (Very poor) to 5 (Very good). Perceived financial burden was evaluated with the item “To what extent do treatment-related expenses affect your financial situation?” Responses were categorized as Not at all, Slightly, Moderately, and Significantly.

All self-report items were administered in the participant’s preferred language (Kazakh or Russian).

### 2.4. Validation Procedure

The validation of the KDQOL-SF™ 1.3 instrument was conducted separately for the Kazakh and Russian versions. The Kazakh version underwent a formal translation and cultural adaptation process following WHO guidelines. This included forward translation by two independent bilingual translators, reconciliation by an expert panel, back-translation into English by a third translator blinded to the original, and review by a multidisciplinary committee of nephrologists, psychologists, and linguists to ensure conceptual and cultural equivalence. The Russian version of the KDQOL-SF™ 1.3 used in this study was a previously validated version developed and psychometrically tested in Russia [18].

Validation procedures in the present study included an assessment of internal consistency reliability using Cronbach’s alpha coefficients for each subscale. In addition, construct validity was examined by calculating Pearson’s correlation coefficients between each subscale score and the global self-rated health item. Significant positive correlations were expected to support the scale’s convergent validity.

Additionally, floor and ceiling effects were examined to assess the instrument’s sensitivity. The percentage of respondents scoring the lowest (floor) or highest (ceiling) possible score on each subscale was calculated. Values exceeding 15% were considered indicative of limited discriminative ability in that domain.

### 2.5. Statistical Analysis

Descriptive statistics were used to summarize participant characteristics and item responses. The internal consistency of the KDQOL-SF™ 1.3 subscales was assessed using Cronbach’s alpha coefficients. Construct validity was examined through Pearson’s correlation between subscale scores and the self-rated overall health item.

The distribution of ESRD etiology by dialysis duration was assessed using a chi-squared test to explore potential group differences. To further evaluate the effects of ESRD etiology, dialysis duration, and their interaction on KDCS, PCS, and MCS, a Bayesian ANOVA was conducted. Model comparisons were performed using Bayes factors (BF_10_) and posterior model probabilities (P(M|data)) to quantify evidence for each model.

Multiple linear regression models were applied to identify predictors of KDCS, PCS, and MCS. Independent variables included sociodemographic factors (age, sex, language, ethnicity, residence, education, and marital and employment status), clinical variables (dialysis duration, ESRD cause, and comorbidities), and patient-reported experiences (access to care, satisfaction, financial burden, and treatment adherence). Variables were dummy-coded where appropriate, with significance set at *p* < 0.05. All statistical analyses were performed using Jamovi software (The jamovi project, version 2.6.17).

### 2.6. Ethical Considerations

The study was conducted in accordance with the ethical principles outlined in the Declaration of Helsinki. Ethical approval was obtained from the Local Bioethics Commission of the “University Medical Center” Corporate Fund (Protocol No. 3 dated 14 July 2023) prior to data collection. All participants were informed about the purpose of the study, the voluntary nature of their participation, and their right to withdraw at any time without consequences. Written informed consent was obtained from participants completing the questionnaire in person, while online participants provided electronic consent prior to beginning the survey. Data were collected and stored anonymously to ensure confidentiality and protect participant privacy.

## 3. Results

A total of 217 hemodialysis patients were included in the study. The mean age was 47.2 years (SD = 13.5), ranging from 20 to 78 years. The sample was nearly evenly distributed by gender, with 50.2% male participants. In terms of age distribution, 34.6% were under 40 years, 45.6% were between 40 and 59 years, and 19.8% were aged 60 years or older (Table 1).

With regard to language, 44.7% of participants completed the questionnaire in Kazakh and 55.3% in Russian. The ethnic composition included 82.5% Kazakh, 11.5% Russian, and 6.0% identifying as other ethnicities. Over half of the sample (50.2%) resided in big cities, followed by 29.5% in cities, 13.8% in towns, and 6.5% in rural areas.

Educational attainment varied, with 40.1% reporting secondary special education and 43.8% having an undergraduate degree. A smaller portion had only middle school (3.2%) or post-graduate education (3.7%). Regarding employment, 36.9% of respondents were receiving disability benefits, while 18.0% were pensioners, 21.7% worked part-time, and only 8.3% were employed full-time. Moreover, 12.4% were unemployed, and 2.8% were students. Most participants were married (61.8%), while 18.0% were single, 12.0% were divorced, and 8.3% were widowed. Moreover, 75.6% reported having children.

In terms of ESRD etiology, the most common cause was glomerulonephritis (30.9%), followed by diabetes mellitus (13.8%), arterial hypertension (10.6%), and polycystic kidney disease (9.2%). Around 16% indicated other causes, and 13.4% were unsure of the underlying cause. Regarding comorbidities, 24.0% reported no comorbid conditions, 35.5% had one, 20.7% had two, and 19.8% had three or more. Among the total study population, the majority of patients were negative for viral hepatitis. Specifically, 33 patients (15.2%) tested positive for at least one hepatitis virus (HBV, HCV, or HDV), whereas 184 patients (84.8%) were negative.

Dialysis duration varied: 18.9% had been on dialysis for less than 1 year, 32.7% for 1–3 years, 22.6% for 4–6 years, 14.7% for 7–10 years, and 11.1% for over 10 years. Across etiologies, the majority of patients had been on dialysis for 1–3 years, with similar distributions in each group (χ^2^ = 27.8, *p* = 0.268; Appendix A).

The mean score for economic well-being was 2.94 (SD = 0.90) on a 5-point scale, indicating moderate financial conditions.

### 3.1. Psychometric Properties of the KDQOL-SF™ 1.3

Both the Kazakh and Russian versions of the KDQOL-SF™ 1.3 demonstrated acceptable to excellent internal consistency across most subscales. Cronbach’s alpha coefficients ranged from 0.751 to 0.877 in the Kazakh version (Table 2) and from 0.692 to 0.933 in the Russian version (Table 3), indicating satisfactory reliability.

In the Kazakh version (*n* = 97), all kidney disease–targeted scales showed Cronbach’s α above 0.75, except for the “Burden of Kidney Disease” scale (α = 0.751) and “Work Status” (α = 0.773). The SF-36 physical and emotional subdomains had strong internal consistency, with Physical Functioning (α = 0.916) and Role–Emotional (α = 0.939) being the highest.

In the Russian version (*n* = 120), internal consistency was similarly robust. The “General Health” subscale showed the lowest reliability (α = 0.692), while the “Physical Functioning” subscale again showed the highest (α = 0.933). Notably, all subscales except Patient Satisfaction had α ≥ 0.70.

Floor and ceiling effects were generally minimal, supporting the instrument’s sensitivity. The Kazakh version showed the highest ceiling effects in the Work Status (35.1%), Cognitive Function (15.5%), and Social Support (20.6%) domains. Floor effects were most prominent for Work Status (46.4%) and Burden of Kidney Disease (23.7%).

In the Russian version, similar patterns emerged, with the highest ceiling effects for Role-Emotional (46.7%), Dialysis Staff Encouragement (25.8%), and Social Support (25.8%), and notable floor effects in Work Status (52.5%), Role-Emotional (32.5%), and Burden of Kidney Disease (18.3%).

Construct validity was examined through correlations between subscale scores and the overall self-rated health measure. In both versions, most kidney disease-specific and generic domains were significantly associated with perceived health status.

In the Kazakh version, the strongest correlations with global health rating were observed for Energy/Fatigue (r = 0.637, *p* < 0.001), General Health (r = 0.730, *p* < 0.001), Cognitive Function (r = 0.494, *p* < 0.001), and Sleep (r = 0.461, *p* < 0.001).

In the Russian version, the top correlates included Energy/Fatigue (r = 0.630, *p* < 0.001), General Health (r = 0.606, *p* < 0.001), Sleep (r = 0.530, *p* < 0.001), and Symptoms/Problems (r = 0.574, *p* < 0.001).

The mean scores for the three main composite scales were as follows (Table 4): Kidney Disease Component Summary (KDCS)—58.1 (SD = 15.0), Physical Component Summary (PCS)—37.5 (SD = 9.8), and Mental Component Summary (MCS)—44.6 (SD = 10.5). The mean self-rated overall health score was 54.8 (SD = 20.6).

All three composite scores were significantly correlated with the overall health rating. The strongest associations were observed for KDCS (r = 0.557, *p* < 0.001) and PCS (r = 0.567, *p* < 0.001), followed by MCS (r = 0.435, *p* < 0.01), supporting the convergent validity of the instrument.

In addition, significant intercorrelations were found among the composite scores themselves: KDCS was highly correlated with both PCS (r = 0.618, *p* < 0.001) and MCS (r = 0.669, *p* < 0.001), while PCS and MCS were modestly correlated (r = 0.190, *p* < 0.01).

### 3.2. Patient-Reported Experiences and Perceptions

Patients reported varied experiences with access, adherence, and satisfaction related to dialysis treatment. Regarding ease of access to the dialysis center, 38.7% described it as somewhat easy, 30.0% as neither easy nor difficult, and 18.9% as somewhat difficult. Only 11.5% found access very easy, while 0.9% (*n* = 2) found it very difficult.

When asked to rate their overall experience with dialysis treatment, nearly half (44.7%) rated it as positive, and 3.2% as very positive. Meanwhile, 36.9% were neutral, and 15.2% rated their experience as negative or very negative.

In terms of treatment adherence, 62.7% of patients reported that they had never missed a dialysis session in the past 3 months, while 28.6% missed 1–2 sessions, and 8.8% missed 3 or more.

Difficulty in following dietary restrictions was common. While 18.9% never experienced difficulty and 17.1% reported it rarely, 40.6% experienced difficulty sometimes, and 23.5% reported it often or always.

Regarding satisfaction with the quality of medical care, 51.6% of patients were satisfied, and 21.2% were very satisfied. An additional 20.3% were somewhat satisfied, while 6.5% were dissatisfied, and 0.5% (*n* = 1) were very dissatisfied.

When asked how treatment-related expenses affected their financial situation, 34.6% reported a moderate impact, and 32.3% reported a significant impact. Only 6.5% indicated that expenses had no impact, while 26.7% said the impact was slight.

### 3.3. Predictors of Health-Related Well-Being

Multiple linear regression analyses were conducted to identify predictors of the three KDQOL-SF™ 1.3 composite scores—Kidney Disease Component Summary (KDCS), Physical Component Summary (PCS), and Mental Component Summary (MCS). The models (Table 5) were statistically significant for all four outcomes: KDCS (F = 13.7, *p* < 0.001, R^2^ = 0.726), PCS (F = 6.3, *p* < 0.001, R^2^ = 0.549), and MCS (F = 4.8, *p* < 0.001, R^2^ = 0.482).

For the KDCS, significant positive predictors included greater satisfaction with dialysis center care (β = 0.247, *p* < 0.001), better economic well-being (β = 0.173, *p* = 0.003), and more positive experience with dialysis treatment (β = 0.180, *p* = 0.002). Compared to patients receiving disability benefits, those who were employed part-time (β = 0.530, *p* < 0.001), employed full-time (β = 0.433, *p* = 0.008), and unemployed (β = 0.316, *p* = 0.029) had significantly higher KDCS scores, indicating better kidney disease-related quality of life among these groups. In contrast, a higher perceived treatment-related financial burden (β = −0.229, *p* < 0.001) was associated with lower KDCS scores. Greater ease of access to the dialysis center was also a significant predictor (β = 0.158, *p* = 0.003).

For the PCS, lower scores were significantly predicted by being female (β = −0.334, *p* = 0.003), more comorbidities (β = −0.188, *p* = 0.016), lower economic well-being (β = 0.277, *p* < 0.001), and higher treatment-related financial burden (β = −0.194, *p* = 0.007).

For the MCS, satisfaction with dialysis center care (β = 0.275, *p* < 0.001), overall dialysis experience (β = 0.189, *p* = 0.015), and lower treatment-related financial burden (β = −0.193, *p* = 0.012) were significant predictors. Employment status was, again, significant.

Moreover, we aimed to assess the effect of ESRD etiology, dialysis duration, and their interaction on KDCS, PCS, and MCS scores using Bayesian ANOVA (Appendix A). For the KDCS, the model including dialysis duration alone showed the highest support (BF_10_ = 1.611). In contrast, for both the PCS and MCS scores, the null model was strongly favored (BF_10_ = 1.000), indicating no substantial effects of ESRD etiology, dialysis duration, or their interaction.

## 4. Discussion

This study aimed to validate the Kazakh and Russian versions of the KDQOL-SF™ 1.3 and examine predictors of health-related quality of life (HRQoL) among hemodialysis patients in Kazakhstan. Although the KDQOL-SF™ 1.3 has been widely validated and utilized in diverse countries worldwide [9,10,12,13,14], its psychometric properties have not yet been assessed in Kazakhstan. The unique linguistic, cultural, and socioeconomic characteristics of Kazakhstan, including the coexistence of Kazakh and Russian languages and disparities in healthcare accessibility, may influence how patients perceive and report their quality of life. Therefore, it is crucial to ensure that both the Kazakh and Russian versions of this tool are not only linguistically accurate but also culturally appropriate and reliable for use in this specific context. Local validation is a necessary step to enable meaningful integration of patient-reported outcomes into routine care and national health monitoring efforts in Kazakhstan. The findings of this study demonstrate that both language versions of the KDQOL-SF™ are psychometrically sound, with acceptable to excellent internal consistency across subscales and strong correlations with global health ratings supporting their construct validity. Additionally, the analysis revealed that HRQoL among dialysis patients is shaped by a combination of clinical, socioeconomic, and experiential factors.

### 4.1. Validity of the KDQOL-SF™ 1.3 in the Kazakhstani Context

The current study provides the first psychometric evaluation of the Kazakh and Russian versions of the KDQOL-SF™ 1.3 among hemodialysis patients in Kazakhstan. Our results confirm that both language versions demonstrate satisfactory psychometric properties, aligning with findings from international validation studies.

Internal consistency, as measured by Cronbach’s alpha, ranged from 0.692 to 0.939 across the subscales, indicating acceptable to excellent reliability. These values are comparable to previous validation studies conducted in Iran [15], Greece [19], and Mexico [20], where most subscales exceeded the 0.70 threshold, and support the robustness of the KDQOL-SF™ in diverse cultural settings. Similar findings were reported in Egypt [16] and Korea [21], where the scale demonstrated strong internal consistency, with lower alpha values primarily limited to subscales such as cognitive function or work status—domains that are known to be culturally sensitive and sometimes context-dependent.

Construct validity in our study was supported by moderate to strong correlations between subscale scores and the global self-rated health item. Domains such as General Health, Energy/Fatigue, and Sleep exhibited the highest associations in both Kazakh and Russian versions, echoing the patterns observed in validations from the Philippines [22], Singapore [23], and Denmark [24]. These results affirm the scale’s convergent validity, particularly for physical and emotional well-being domains, which are central to patients’ lived experiences on dialysis.

Floor and ceiling effects were also evaluated to assess sensitivity. Consistent with prior studies, including those from Iran [15] and India [25], the KDQOL-SF™ demonstrated acceptable distribution properties, with limited concentration at score extremes. This suggests that the instrument retains its discriminative ability in the Kazakhstani context, avoiding saturation in patient responses and thus maintaining its utility for clinical and research applications.

Taken together, our results support the cultural equivalence, reliability, and construct validity of the KDQOL-SF™ 1.3 in Kazakhstan. Importantly, the dual-language administration approach, aligned with best practices from previous transcultural adaptations [19,22,23], ensured inclusivity and linguistic appropriateness, enhancing the instrument’s feasibility for use in multiethnic populations.

### 4.2. Predictors of Quality of Life Among Hemodialysis Patients

Our findings highlight that access to dialysis centers remains a challenge for a substantial portion of patients. While 38.7% found it somewhat easy to reach the facility, 30.0% were neutral and 18.9% reported some difficulty; only 11.5% described access as very easy, and 0.9% as very difficult. These results are consistent with existing evidence showing that travel-related burdens can negatively affect dialysis adherence and health outcomes. Prior studies have linked longer travel distances with reduced session attendance, poorer quality of life, and increased mortality. Improving local access through satellite units or transport support may help address these challenges and enhance patient experience and well-being [26].

Patient satisfaction with dialysis care showed mixed results in our study. While nearly one-half (44.7%) rated their experience as positive and 3.2% as very positive, over one-third (36.9%) were neutral, and 15.2% reported negative or very negative experiences. These findings reflect the broader literature, where patient satisfaction is recognized as a key indicator of care quality and linked to adherence and outcomes [27,28,29]. The presence of neutral or negative ratings in a sizable portion of our sample highlights opportunities to enhance patient-centered care, particularly in aspects such as communication, environment, and support during treatment.

Our findings indicate that while most patients (62.7%) did not miss any dialysis sessions in the past three months, a notable proportion reported non-adherence, with 28.6% missing 1–2 sessions and 8.8% missing 3 or more. Adherence to dietary restrictions also proved challenging: only 18.9% reported never experiencing difficulty, while 64.1% faced challenges at least sometimes. These patterns reflect the complexity of maintaining strict dialysis regimens and highlight the need for supportive strategies—such as dietary counseling and patient education—to address barriers to adherence and promote better health outcomes [30].

A substantial proportion of patients reported experiencing financial strain due to dialysis-related expenses. Specifically, 34.6% indicated a moderate impact and 32.3% a significant impact on their financial situation, while only 6.5% reported no impact. These findings highlight the economic burden associated with dialysis care, which may contribute to stress and reduced well-being. Addressing financial challenges through support measures, such as subsidies or transport assistance, may help reduce this burden and improve the overall patient experience [31,32].

Our results confirm that health-related quality of life (HRQoL) in dialysis patients is influenced by clinical, socioeconomic, and care-related factors. Using the KDQOL-SF™ 1.3 instrument, we found that greater satisfaction with dialysis care, higher economic well-being, and a more positive treatment experience were associated with better kidney-specific quality of life (KDCS). In contrast, a higher treatment-related financial burden was a strong negative predictor (*p* < 0.001). Employment status also mattered: patients employed part-time or full-time, as well as unemployed individuals, had significantly higher KDCS scores than those receiving disability benefits—suggesting potential psychosocial or functional differences between these groups [33]. Easier access to dialysis centers also predicted higher KDCS scores.

Lower physical health (PCS) was associated with being female, having more comorbidities, and experiencing financial hardship. These findings reflect known challenges faced by women and those with multiple chronic conditions [34]. Conversely, better economic well-being and care experience were linked to higher PCS scores.

Mental health (MCS) was positively associated with satisfaction with care and dialysis experience, and negatively associated with financial burden. Employment status also predicted MCS, aligning with literature showing the mental health benefits of vocational engagement and social roles.

Together, these findings highlight the multidimensional influences on HRQoL in dialysis patients. Financial strain, comorbidities, and care experiences affect not only physical functioning but also psychological well-being. The consistent impact of economic hardship across KDCS, PCS, and MCS supports the concept of “financial toxicity” in chronic disease. Patient-centered strategies (e.g., improving care quality, communication, and support) may enhance both clinical and psychosocial outcomes.

### 4.3. Policy and Clinical Practice Implications

The findings of this study offer actionable insights for both policy development and clinical practice improvement in Kazakhstani hemodialysis services and similar contexts. First, the successful validation of the KDQOL-SF™ 1.3 provides a reliable, culturally appropriate tool for the routine assessment of patient-reported outcomes in both Kazakh and Russian. Integrating this tool into national quality monitoring systems and regular clinical practice can help clinicians better understand patients’ physical, emotional, and social well-being, enabling more personalized and holistic care planning beyond standard biomedical indicators.

Second, our results underscore the critical role of economic and psychosocial factors in shaping patient experiences and outcomes. Financial burden emerged as a strong negative predictor of both physical and mental health components, highlighting the need for broader social and financial support mechanisms. Policies aimed at reducing indirect costs, such as subsidies for transportation, medications, or financial counseling services, could meaningfully alleviate the economic strain and improve adherence and overall quality of life.

Moreover, patient satisfaction and positive dialysis experiences were strongly associated with better HRQoL. This emphasizes the importance of enhancing relational skills and communication among healthcare providers, particularly nephrology nurses and dialysis staff. Training programs focused on empathetic communication, shared decision-making, and emotional support should be integrated into routine practice to strengthen therapeutic relationships and build patient trust.

Finally, given the high prevalence of comorbidities and challenges in physical functioning among hemodialysis patients, integrating multidisciplinary care approaches, such as including dietitians, social workers, mental health professionals, and physiotherapists, can help address patients’ multifaceted needs. Implementing such comprehensive, patient-centered strategies can not only improve health-related quality of life but also lead to better long-term clinical outcomes and more efficient health system utilization.

Overall, policies and clinical practices that reduce structural barriers (such as distance and cost), enhance psychosocial and economic support, and promote patient-centered care will not only improve patient-reported well-being but also potentially improve survival and reduce healthcare burdens. In a context where the dialysis population continues to grow globally, these integrated approaches are essential to achieving meaningful improvements in quality of life and health outcomes for individuals with ESRD [31,35].

### 4.4. Study Limitations

This study has several limitations. First, the cross-sectional design limits the ability to draw causal inferences regarding the directionality of observed associations. The relationships identified in this study should be interpreted as associations rather than causal effects, and we encourage future research to investigate these findings longitudinally. Second, self-reported data on dialysis adherence, financial burden, and treatment satisfaction may be subject to recall bias or social desirability bias. Third, although the sample included diverse linguistic and geographic groups, it was recruited via convenience sampling, which may limit generalizability to the entire dialysis population in Kazakhstan.

Despite these limitations, the study provides valuable foundational evidence on HRQoL among hemodialysis patients in Kazakhstan, where such data had previously been lacking. We believe that our findings offer an important starting point for further research and can inform improvements in patient-centered dialysis care in this context.

### 4.5. Future Perspectives

Building on this study, future research should focus on longitudinal assessments of HRQoL to examine how individual and systemic changes affect outcomes over time. Incorporating the KDQOL-SF™ 1.3 into routine clinical practice could allow for real-time tracking of quality-of-life trends and help guide personalized care planning.

In addition, future studies should explore the integration of HRQoL metrics into national dialysis registries, enabling policy-relevant population-level monitoring. Further validation of the scale in pediatric or peritoneal dialysis populations, as well as in other Central Asian contexts, would enhance its applicability. Lastly, mixed-methods approaches combining qualitative interviews with quantitative assessments could deepen the understanding of cultural, social, and emotional factors influencing HRQoL in this setting.

## 5. Conclusions

This study provides the first evidence supporting the reliability and validity of the Kazakh and Russian versions of the KDQOL-SF™ 1.3 for assessing health-related quality of life among hemodialysis patients in Kazakhstan. The instrument demonstrated acceptable internal consistency and construct validity across both language versions, affirming its suitability for routine use in a multilingual clinical context.

Beyond psychometric validation, the study identified key social, economic, and care-related predictors of quality of life. Higher satisfaction with medical care, better economic well-being, and positive dialysis experiences were consistently associated with improved HRQoL, while financial burden and limited access to care were linked to poorer outcomes. These findings underline the need for integrated, patient-centered approaches that consider not only clinical indicators but also economic and psychosocial factors affecting patients’ lives.

Importantly, the integration of validated patient-reported outcome measures such as the KDQOL-SF™ into national quality monitoring systems and routine clinical practice has the potential to facilitate more personalized, data-driven care strategies. Addressing socioeconomic disparities and strengthening psychosocial support services should be prioritized as part of comprehensive CKD management in Kazakhstan and similar settings.

Despite the study’s cross-sectional nature, these results provide a strong foundation for future longitudinal research, which is essential to better understand causal pathways and long-term trends in HRQoL among dialysis patients. Continued efforts to integrate new technologies, enhance the relational skills of dialysis staff, and promote multidisciplinary care are crucial for achieving substantial improvements in patient outcomes and overall health system efficiency.

## Figures and Tables

**Table 1 jcm-14-05021-t001:** Study population (*N* = 217).

Variable	*n*	%
Gender		
Male	109	50.2
Female	108	49.8
Age group		
<40 years	75	34.6
40–59 years	99	45.6
≥60 years	43	19.8
Language		
Kazakh	97	44.7
Russian	120	55.3
Ethnicity		
Kazakh	179	82.5
Russian	25	11.5
Other	13	6.0
Residence		
Rural	14	6.5
Town	30	13.8
City	64	29.5
Big city	109	50.2
Educational level		
Middle school	7	3.2
High school	20	9.2
Secondary special	87	40.1
Undergraduate	95	43.8
Post-graduate	8	3.7
Occupation		
Full-time employment	18	8.3
Part-time employment	47	21.7
Unemployed	27	12.4
Pensioner	39	18.0
On disability benefits	80	36.9
Student	6	2.8
Marital status		
Single	39	18.0
Married	134	61.8
Divorced	26	12.0
Widow	18	8.3
Children		
No	53	24.4
Yes	164	75.6
ESRD ^1^ etiology		
Glomerulonephritis	67	30.9
Diabetes mellitus	30	13.8
Arterial hypertension	23	10.6
Polycystic kidney disease	20	9.2
Pyelonephritis	14	6.5
Other	34	15.7
Don’t know	29	13.4
Dialysis duration		
<1 year	41	18.9
1–3 years	71	32.7
4–6 years	49	22.6
7–10 years	32	14.7
>10 years	24	11.1
Comorbid diseases		
No	52	24.0
1	77	35.5
2	45	20.7
≥3	43	19.8
Positive viral hepatitis status (B/C/D)		
No	184	84.8
Yes	33	15.2

^1^ ESRD—End-stage renal disease.

**Table 2 jcm-14-05021-t002:** Validity of Kazakh version of the KDQOL-SF™ 1.3 among hemodialysis patients in Kazakhstan (*n* = 97).

Scales	M (SD) ^1^	Ceiling %	Floor %	Cronbach’s α	Overall Health Rate, r (*p*)
Kidney disease targeted scales					
Symptoms/problems	68.4 (16.2)	0	0	0.869	0.362 (<0.001)
Effects of kidney disease	53.8 (24.0)	4.1	0	0.877	0.009 (0.931)
Burden of kidney disease	23.1 (20.1)	0	23.7	0.751	0.398 (<0.001)
Work status	44.3 (45.0)	35.1	46.4	0.773	0.242 (0.017)
Cognitive function	72.2 (21.4)	15.5	0	0.804	0.494 (<0.001)
Quality of social interaction	65.6 (23.6)	3.1	0	0.800	0.453 (<0.001)
Sleep	60.5 (18.0)	0	0	0.790	0.461 (<0.001)
Social support	67.7 (21.1)	20.6	0	0.814	0.433 (<0.001)
Dialysis staff encouragement	69.1 (22.2)	17.5	0	0.857	0.175 (0.086)
Patient satisfaction	52.2 (23.5)	7.2	1.0	-	0.260 (0.010)
SF-36 ^2^					
Physical functioning	55.6 (26.8)	2.1	4.1	0.916	0.425 (<0.001)
Role—physical	36.6 (42.2)	25.8	49.5	0.896	0.383 (<0.001)
Pain	59.4 (18.9)	5.2	0	0.679	0.417 (<0.001)
General health	32.2 (21.9)	0	6.2	0.822	0.730 (<0.001)
Emotional well-being	59.8 (19.5)	0	0	0.798	0.497 (<0.001)
Role—emotional	65.3 (45.1)	60.8	27.8	0.939	0.214 (0.035)
Social function	65.6 (21.5)	12.4	0	0.843	0.394 (<0.001)
Energy/fatigue	42.6 (22.0)	0	0	0.808	0.637 (<0.001)

^1^ M (SD)—Mean (Standard Deviation). ^2^ SF-36—Short Form 36.

**Table 3 jcm-14-05021-t003:** Validity of Russian version of the KDQOL-SF™ 1.3 among hemodialysis patients in Kazakhstan (*n* = 120).

Scales	M (SD) ^1^	Ceiling %	Floor %	Cronbach’s α	Overall Health Rate, r (*p*)
Kidney disease targeted scales					
Symptoms/problems	71.2 (19.2)	0.8	0	0.891	0.574 (<0.001)
Effects of kidney disease	53.6 (27.1)	4.2	1.7	0.905	0.248 (0.006)
Burden of kidney disease	27.4 (22.9)	1.7	18.3	0.785	0.349 (<0.001)
Work status	36.3 (42.0)	0.25	52.5	0.722	0.340 (<0.001)
Cognitive function	75.0 (21.2)	17.5	0.8	0.853	0.467 (<0.001)
Quality of social interaction	67.2 (21.7)	5.0	0	0.726	0.424 (<0.001)
Sleep	57.6 (18.1)	0	0	0.761	0.530 (<0.001)
Social support	66.2 (27.0)	25.8	2.5	0.735	0.258 (0.004)
Dialysis staff encouragement	72.2 (24.0)	25.8	2.5	0.708	0.293 (0.001)
Patient satisfaction	58.2 (23.4)	10.0	0.8	-	0.360 (<0.001)
SF-36 ^2^					
Physical functioning	54.3 (30.5)	1.7	10.0	0.933	0.332 (<0.001)
Role—physical	40.2 (42.9)	25.8	46.7	0.896	0.363 (<0.001)
Pain	59.2 (24.1)	12.5	1.7	0.793	0.537 (<0.001)
General health	42.6 (21.8)	1.7	4.2	0.692	0.606 (<0.001)
Emotional well-being	56.8 (19.0)	0	0	0.687	0.403 (<0.001)
Role—emotional	55.3 (45.0)	46.7	32.5	0.889	0.202 (0.016)
Social function	63.8 (24.9)	16.7	1.7	0.729	0.488 (<0.001)
Energy/fatigue	52.0 (20.9)	0.8	0.8	0.694	0.630 (<0.001)

^1^ M (SD)—Mean (Standard Deviation). ^2^ SF-36—Short Form 36.

**Table 4 jcm-14-05021-t004:** Correlation between the main composite summary scores and overall health rate of the KDQOL-SF™ 1.3 among hemodialysis patients in Kazakhstan (*N* = 217).

Scale	M (SD)	(1)	(2)	(3)	(4)
KDCS ^1^ (1)	58.1 (15.0)	-			
PCS ^2^ (2)	37.5 (9.80)	0.618 ***	-		
MCS ^3^ (3)	44.6 (10.5)	0.669 ***	0.190 **	-	
Overall health rate (4)	54.8 (20.6)	0.557 ***	0.567 ***	0.435 **	-

^1^ KDCS—Kidney Disease Component Summary. ^2^ PCS—Physical Component Summary. ^3^ MCS—Mental Component Summary. Note: ** *p* < 0.01; *** *p* < 0.001.

**Table 5 jcm-14-05021-t005:** Predictors of main composite summary scores and overall health rate of the KDQOL-SF™ 1.3 among hemodialysis patients in Kazakhstan (*N* = 217).

Model	KDCS ^1^		PCS ^2^		MCS ^3^	
	F = 13.7, *p* < 0.001 R^2^ = 0.726		F = 6.3, *p* < 0.001 R^2^ = 0.549		F = 4.8, *p* < 0.001 R^2^ = 0.482	
Predictors	β	*p*	β	*p*	β	*p*
Gender						
Female–Male	−0.126	0.146	−0.334	0.003	0.034	0.776
Age						
40–59−<40 years	0.026	0.816	0.100	0.494	0.289	0.066
≥60−<40 years	−0.099	0.689	−0.128	0.686	0.009	0.979
Language						
Kazakh–Russian	0.024	0.813	−0.160	0.211	0.287	0.038
Ethnicity						
Russian–Kazakh	−0.032	0.821	−0.270	0.142	0.186	0.343
Other–Kazakh	−0.064	0.741	−0.138	0.578	0.098	0.712
Residence						
Rural–Big city	0.142	0.489	−0.143	0.587	0.075	0.790
Town–Big city	0.072	0.615	−0.174	0.343	0.087	0.659
City–Big city	0.004	0.969	0.026	0.842	0.043	0.759
Educational level	0.023	0.631	−0.036	0.566	0.007	0.915
Occupation						
Student–On disability benefits	0.417	0.135	0.515	0.150	0.538	0.161
Pensioner–On disability benefits	0.037	0.863	−0.498	0.073	0.743	0.013
Unemployed–On disability benefits	0.316	0.029	−0.048	0.794	0.444	0.025
PT ^4^-employment–On disability benefits	0.530	<0.001	−0.061	0.712	0.479	0.007
FT ^5^-employment–On disability benefits	0.433	0.008	−0.146	0.485	0.401	0.075
Marital status						
Married–Single	0.185	0.219	0.019	0.921	−0.010	0.961
Divorced–Single	0.430	0.021	0.170	0.475	0.116	0.649
Widow–Single	0.272	0.228	0.147	0.612	−0.187	0.546
Children						
Yes–No	−0.067	0.660	−0.023	0.906	−0.005	0.981
ESRD ^6^ etiology						
Glomerulonephritis–Don’t know	−0.260	0.084	−0.113	0.557	−0.060	0.771
Diabetes mellitus–Don’t know	−0.293	0.095	−0.072	0.748	0.180	0.455
Arterial hypertension–Don’t know	0.102	0.565	0.004	0.985	0.047	0.847
PKD ^7^–Don’t know	−0.343	0.074	−0.159	0.517	−0.228	0.386
Pyelonephritis–Don’t know	0.041	0.844	−0.116	0.663	−0.259	0.365
Other–Don’t know	0.067	0.685	−0.230	0.278	−0.042	0.852
Dialysis duration	0.048	0.314	0.045	0.457	0.053	0.127
Comorbidities	−0.017	0.777	−0.188	0.016	0.053	0.527
Positive viral hepatitis status (B/C/D)	−0.171	0.187	0.116	0.484	−0.337	0.059
Ease of access to the dialysis center	0.158	0.003	0.084	0.222	0.002	0.977
Experience with dialysis treatment	0.180	0.002	0.141	0.052	0.189	0.015
Frequency of missed dialysis sessions	−0.001	0.907	0.054	0.373	0.013	0.835
Difficulties in following dietary restrictions	−0.018	0.689	−0.002	0.972	−0.020	0.745
Satisfaction with dialysis centers’ care	0.247	<0.001	−0.058	0.422	0.275	<0.001
Economic well-being	0.173	0.003	0.277	<0.001	0.148	0.062
Effects of treatment-related expenses	−0.229	<0.001	−0.194	0.007	−0.193	0.012

^1^ KDCS—Kidney Disease Component Summary. ^2^ PCS—Physical Component Summary. ^3^ MCS—Mental Component Summary. ^4^ PT—part-time employment. ^5^ FT—Full-time employment. ^6^ ESRD—End-stage renal disease. ^7^ PKD—Polycystic kidney disease.

## Data Availability

The dataset supporting the conclusions of this article is available from the corresponding author on reasonable request.

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
