# Peer review of "Assessing Quality of Life in Hemodialysis Patients in Kazakhstan: A Cross-Sectional Study"

_jcm, 2025, doi:10.3390/jcm14145021_

Round 1

Reviewer 1 Report

Comments and Suggestions for Authors

Review for the Journal of clinical medicine on the article:

 “When Organ Donation Is Rare, Every Measure Counts: Validating the KDQOL-SFTM 1.3 and Exploring Quality of Life on Hemodialysis in Kazakhstan”

Aruzhan Asanova 1†, Aidos Bolatov 1,2*†, Deniza Suleimenova 3,4, Yelnur Khazhgaliyeva 1 , Saule Shaisultanova 5, Sholpan Altynova 5 , Yuriy Pya 6

The authors conducted a cross sectional survey on dialysis patients in Kazakhstan where access to kidney transplantation is a real challenge from several points of view. The authors provide an analysis that tries to validate the Russian and Kazakh version of the KDQUL-SFTM 1.3 and to examine predictors of HRQoL among hemodialysis (HD) patients in Kazakhstan.

The statistical and methodological approach that the authors used to validate the tools in this population was adequate and well conducted, but the study has in my opinion several flaws.

First, the authors validate a tool that was already validate almost worldwide, therefore this is important in a context that could not rely on a solid healthcare system, but it’s not a novelty.

Second, as the authors describe, “the cross-sectional design limits the ability to draw causal inferences regarding the directionality of observed associations” therefore it could be published in a journal with a lower IF because it’s difficult to drive conclusion on this matter.

Third, the Quality of life analysis is a very important tool but it should be examinated in a long prospective, it should consider the starting point of every patient. How would they answer to several questions concerning their quality of life at the beginning of HD treatment? How would they answer after 5 or 10 years? A longitudinal study would be more suitable for such an analysis in my opinion. Last, thus there are only a few studies exploring HD population in Kazakhstan, the sample examinated is quite small to drive conclusion on the matter.

Author Response

We sincerely thank the reviewer for taking the time to review our manuscript and for providing thoughtful and constructive comments. We greatly appreciate your suggestions, which have helped us further improve the clarity and quality of our work.

Comment 1: The statistical and methodological approach that the authors used to validate the tools in this population was adequate and well conducted, but the study has in my opinion several flaws.

Response 1: We sincerely thank the reviewer for their positive evaluation of our statistical and methodological approach, and for acknowledging that these aspects were well conducted. We also appreciate the reviewer’s honest assessment regarding the overall study and understand that further clarification and improvement were needed to address the identified flaws.

In the following responses, we have carefully addressed each specific concern raised and have revised the manuscript accordingly to improve its clarity, completeness, and scientific rigor.

Comment 2: First, the authors validate a tool that was already validate almost worldwide, therefore this is important in a context that could not rely on a solid healthcare system, but it’s not a novelty.

Response 2: We thank the reviewer for this important observation. We agree that the KDQOL-SFTM 1.3 has been widely validated in various countries and settings. However, to our knowledge, no prior validation has been conducted in Kazakhstan, where unique cultural, linguistic, and socioeconomic factors may influence patients’ health-related quality of life (HRQoL) perceptions and responses to the instrument.

Furthermore, Kazakhstan has a multilingual healthcare environment (Kazakh and Russian), and it was essential to ensure that both language versions of the KDQOL-SFTM 1.3 were psychometrically sound and culturally appropriate for local use. This step is critical for implementing patient-reported outcome measures in routine clinical practice and national quality monitoring in Kazakhstan.

We have clarified this point in the Discussion section (first paragraph) to emphasize the regional relevance and practical significance of our validation work, even if it may not represent a global novelty.

Comment 3: Second, as the authors describe, “the cross-sectional design limits the ability to draw causal inferences regarding the directionality of observed associations” therefore it could be published in a journal with a lower IF because it’s difficult to drive conclusion on this matter.

Response 3: We thank the reviewer for highlighting this important limitation and for their critical perspective regarding the level of evidence that can be drawn from a cross-sectional design. We fully acknowledge that the cross-sectional nature of our study does not allow for causal inferences, and we have clearly stated this limitation in the Discussion section.

However, we respectfully note that cross-sectional studies remain a valuable first step for generating evidence in under-researched settings, particularly when foundational data on patient-reported outcomes are lacking. To date, there had been no psychometric validation of the KDQOL-SFTM 1.3 in Kazakhstan, nor data on HRQoL predictors in this population. Our study provides essential baseline evidence needed to inform future longitudinal studies and policy interventions.

We have further emphasized in the revised Discussion that our findings should be interpreted as exploratory and hypothesis-generating, and we encourage future longitudinal research to confirm and expand upon these associations. We believe that this foundational work makes a meaningful contribution to the literature on dialysis care in Central Asia and supports the global discourse on patient-reported outcomes.

Comment 4: Third, the Quality of life analysis is a very important tool but it should be examinated in a long prospective, it should consider the starting point of every patient. How would they answer to several questions concerning their quality of life at the beginning of HD treatment? How would they answer after 5 or 10 years? A longitudinal study would be more suitable for such an analysis in my opinion. Last, thus there are only a few studies exploring HD population in Kazakhstan, the sample examinated is quite small to drive conclusion on the matter.

Response 4: We thank the reviewer for this insightful comment highlighting the importance of longitudinal assessment in evaluating changes in quality of life over time. We fully agree that a prospective design, starting from the initiation of hemodialysis (HD) and following patients over several years, would provide valuable insights into the dynamics and trajectories of HRQoL.

As noted in our revised manuscript (see revised Sections 4.3. and 4.4), we explicitly acknowledge this limitation and emphasize the need for future longitudinal studies to explore changes in HRQoL over time, including assessments at treatment initiation and at different follow-up intervals.

Regarding the sample size, we recognize that while our study included 217 participants, which is comparable to or larger than several previous validation studies in similar contexts, it remains relatively modest. We have clarified in the manuscript that our findings should be interpreted cautiously and primarily as exploratory and hypothesis-generating. Our work provides a first essential step toward understanding HRQoL among HD patients in Kazakhstan and can serve as a foundation for larger, longitudinal, and multicenter studies in the future.

We have added text in the Discussion to emphasize these points and to encourage future research efforts that build upon our initial findings.

Reviewer 2 Report

Comments and Suggestions for Authors

recommendations

  1. The title better to be more short
  2. Better to place the following above the aim. By addressing these gaps, this study provides the first evidence on the applicability 104 of the KDQOL-SFTM in Kazakhstan and contributes to a better understanding of factors 105 influencing patient-reported outcomes in a multilingual dialysis context.
  3. line 116-117 is there any bias when using 2 methods? were all subjects under the same circumstances of measurement?
  4. line 120-122 : belong to ethical
  5. line 123: replace with the word research instrument
  6. table 1 : delete the word student
  7. separate future perspectives from study limitations
  8. Please replace the references in which you are includesd:
    Bolatov, A., Asanova, A., Daniyarova, G., Sazonov, V., Semenova, Y., Abdiorazova, A., & Pya, Y. (2025). Barriers and willingness 516 to express consent to organ donation among the Kazakhstani population. BMC public health, 25(1), 842. 517 https://doi.org/10.1186/s12889-025-22044-4 518 3.Bolatov, A., Asanova, A., Abdiorazova, A., & Pya, Y. (2025). Too uncertain to consent, too supportive to refuse: The sociocultural 519 dilemma of hesitant organ donors in Kazakhstan. Frontiers in Public Health, 13, 1602008. 520 https://doi.org/10.3389/fpubh.2025.1602008

Author Response

We sincerely thank the reviewer for taking the time to review our manuscript and for providing thoughtful and constructive comments. We greatly appreciate your suggestions, which have helped us further improve the clarity and quality of our work.

Comment 1: The title better to be more short

Response 1: We thank the reviewer for this suggestion. In response, we have revised the title to make it more concise while retaining key information about the study context and population. The new title is:

“Assessing Quality of Life in Hemodialysis Patients in Kazakhstan: A Cross-Sectional Study”

We believe this revised title is now clearer and more focused.

Comment 2: Better to place the following above the aim. By addressing these gaps, this study provides the first evidence on the applicability 104 of the KDQOL-SFTM in Kazakhstan and contributes to a better understanding of factors 105 influencing patient-reported outcomes in a multilingual dialysis context.

Response 2: We thank the reviewer for this helpful suggestion regarding the placement of this sentence. We agree that moving this contextual statement before the aims can improve the logical flow and better introduce the purpose of the study.

Comment 3: line 116-117 is there any bias when using 2 methods? were all subjects under the same circumstances of measurement?

Response 3: We thank the reviewer for raising this important point regarding potential bias from using two data collection methods. We agree that differences in administration mode could introduce bias.

In our study, all participants received the same standardized questionnaire content and instructions, regardless of the mode (in-person or online). Additionally, both methods were designed to minimize interviewer influence and to ensure privacy. While some degree of bias (e.g., related to self-selection in online participation) cannot be completely ruled out, we believe that using both methods allowed us to reach a more diverse sample, including patients from different geographic and social backgrounds.

Comment 4: line 120-122 : belong to ethical

Response 4: We thank the reviewer for this helpful suggestion. We agree that this sentence fits better within the Ethical Considerations section. Accordingly, we have deleted this sentence from the Methods section and ensured that information regarding informed consent is clearly included in the Ethical Considerations section instead.

Comment 5: line 123: replace with the word research instrument

Response 5: We thank the reviewer for this precise suggestion. We have revised the text as recommended and replaced the term with “research instrument”.

Comment 6: table 1 : delete the word student

Response 6: We reviewed the table carefully and corrected a minor typo in formatting to ensure clarity.

Comment 7: separate future perspectives from study limitations

Response 7: We thank the reviewer for this valuable suggestion. In line with your recommendation, we have separated the “Future Perspectives” into a distinct section, independent from “Study Limitations,” to improve clarity and readability. The manuscript now clearly presents these as two separate sections (Sections 4.3 and 4.4).

Comment 8: Please replace the references in which you are includesd:

Bolatov, A., Asanova, A., Daniyarova, G., Sazonov, V., Semenova, Y., Abdiorazova, A., & Pya, Y. (2025). Barriers and willingness 516 to express consent to organ donation among the Kazakhstani population. BMC public health, 25(1), 842. 517 https://doi.org/10.1186/s12889-025-22044-4 518

Bolatov, A., Asanova, A., Abdiorazova, A., & Pya, Y. (2025). Too uncertain to consent, too supportive to refuse: The sociocultural 519 dilemma of hesitant organ donors in Kazakhstan. Frontiers in Public Health, 13, 1602008. 520 https://doi.org/10.3389/fpubh.2025.1602008

Response 8: We thank the reviewer for this important comment. As recommended, we have removed the references in which we are included from the manuscript to avoid potential self-citation bias.

Reviewer 3 Report

Comments and Suggestions for Authors

Dear Author,

see the comments in the annex file.

Best

Author Response

Comment 1: First of all, I would like to express my sincere gratitude for the opportunity to contribute my opinion to the evaluation of your manuscript. I found the topic both interesting and relevant to the field in which we operate. Below, I list the main areas that could benefit from further elaboration and revision.

Response 1: We sincerely thank the reviewer for the time and effort dedicated to evaluating our manuscript, as well as for the thoughtful and constructive feedback provided. We truly appreciate your positive remarks regarding the relevance and importance of our topic.

Below, we provide detailed point-by-point responses to each of your suggestions. We have carefully addressed all comments and made corresponding revisions in the manuscript to improve its clarity, depth, and overall quality.

Comment 2: Title: Acceptable, but I advise against the use of acronyms in this section (as well as in the keywords section).

Response 2: We thank the reviewer for this helpful comment. In response, we have revised the title to remove acronyms entirely. The new title is:

“Assessing Quality of Life in Hemodialysis Patients in Kazakhstan: A Cross-Sectional Study”

We have updated the keywords to remove acronyms and ensure they are concise and focused on the study’s objectives, outcomes, and setting. The revised keywords are:

“hemodialysis; quality of life; validation; predictors; Kazakhstan.”

We believe this change improves clarity and aligns with your recommendation.

Comment 3: Editing: The references in the dedicated section do not follow the journal’s and the editor’s required template. The proposed tables need legends explaining the acronyms used (e.g., SD, SF-36…).

Response 3: We thank the reviewer for carefully noting these important editorial points. We have revised the reference list to fully comply with the journal’s formatting requirements. Additionally, we have updated all table legends to include explanations of the acronyms used.

Comment 4: Abstract: I suggest expanding the section related to results and conclusions, especially from a clinical practice perspective (see Introduction and Discussion).

Response 4: We thank the reviewer for this valuable suggestion. In response, we have expanded the Abstract to provide more detailed results, including the influence of socioeconomic and access-related factors on HRQoL, and emphasized the clinical practice implications of our findings. The revised Abstract now highlights the importance of integrating patient-reported outcome measures into routine dialysis care and addressing economic and access barriers to improve patient-centered outcomes.

Comment 5: Keywords: Overall acceptable, but I recommend using a maximum of 4–5 terms focused on the study objectives, outcomes, study type, and setting. The first two terms are redundant. Also, the use of acronyms is discouraged in this section.

Response 5: We thank the reviewer for this helpful suggestion. We have revised the keywords to include a maximum of five focused terms that better reflect the study objectives, outcomes, and setting, and we have removed acronyms as recommended. The revised keywords are now: Hemodialysis; quality of life; validation; predictors; Kazakhstan.

Comment 6: Introduction: This section lacks clinical practice perspectives (see Abstract and Discussion) and international context. Additionally, the manuscript is missing content related to the care and management of CKD in the pre-transplant phase (ESRD), which represents the sample population studied. In this regard, I suggest expanding the discussion by citing recent scientific literature to support the following topics: "Relational skills of nephrology and dialysis nurses in clinical care settings", "New technology in dialysis management", and "Updates in chronic kidney disease management". These would enrich the introduction and broaden the audience of readers and researchers interested in the dissemination of your findings.

Response 6: We thank the reviewer for this valuable and insightful comment. In response, we have substantially expanded the Introduction to include a broader clinical practice perspective and provide an updated international context. Specifically, we added discussion on the importance of relational skills and communication competencies of nephrology nurses, recent technological innovations in dialysis management, and updates in chronic kidney disease (CKD) management in the pre-transplant phase.

The new text emphasizes the shift toward patient-centered and holistic approaches in CKD care and highlights the relevance of integrating validated HRQoL tools in this context. Relevant recent scientific literature has been cited to support these additions.

Comment 7: The objectives are generally acceptable; I only suggest using the classic format: “The primary objective was… whereas the secondary objective was…”. It might also be beneficial to include the specific research questions the authors intend to answer—perhaps dedicating a separate section in the Methods (e.g., 2.1 “Aims and Research Questions”).

Response 7: We thank the reviewer for this helpful suggestion. In response, we have revised the presentation of our objectives to clearly distinguish between the primary and secondary objectives using the classic phrasing. Additionally, we have included specific research questions to clarify the focus of our study. To improve clarity and readability, we have dedicated a new subsection titled “2.1 Aims and Research Questions” in the Methods section where this information is now presented.

Comment 8: This is perhaps the weakest part and certainly deserves more attention, as some key elements are missing or insufficiently described. The reporting guideline used is not mentioned, nor is the corresponding checklist included in the supplementary files, nor is it cited among the references (e.g., EQUATOR Network: https://www.equator-network.org/ or STROBE: https://www.strobe-statement.org/, depending on the authors’ choice). Following a reporting tool would help the authors achieve greater clarity in this crucial section and is mandatory for the relevant scientific community.

Response 8: We thank the reviewer for highlighting this important point. In response, we have revised our manuscript to explicitly state that our study was conducted and reported in accordance with the Strengthening the Reporting of Observational Studies in Epidemiology (STROBE) guidelines. We have included a citation to the STROBE statement in the references and added the completed STROBE checklist as a supplementary file, as recommended. We believe this addition improves the clarity, transparency, and rigor of our Methods and overall reporting.

Comment 9: Results: Overall, this section is well done. It is the strength of the current version of the manuscript and will benefit further from the previous and following suggestions. Only the table formatting needs improvement, as mentioned earlier.

Response 9: We thank the reviewer for the positive feedback on the Results section and for recognizing it as a strength of our manuscript. In line with your suggestion, we have carefully revised and improved the formatting of all tables to ensure clarity and consistency. We appreciate your helpful comments, which have contributed to enhancing the overall quality and presentation of our findings.

Comment 10: Discussion: Generally well-written but insufficiently explores the “Perspectives for Clinical Practice” section, which (see Abstract and Introduction) could broaden the audience of this work. I recommend expanding this part, either by creating a dedicated section titled "Perspectives for Clinical Practice", or by broadening the existing "Policy Recommendations" section, possibly covering broader disease management topics. These suggestions align with the advice given in the Introduction and would undoubtedly attract more readers and researchers interested in developing a broader critical understanding.

Response 10: We thank the reviewer for this insightful and constructive suggestion. In response, we have substantially expanded the discussion on clinical practice implications and integrated it with the policy recommendations into a new combined subsection titled “Policy and Clinical Practice Implications.” This revised section addresses broader topics, including economic and psychosocial support, communication skills and training for dialysis staff, and the importance of multidisciplinary, patient-centered care approaches. We believe this addition not only strengthens the discussion but also broadens the relevance and appeal of our findings to clinicians, policymakers, and researchers interested in improving dialysis care.

Comment 11: Limitations: Generally acceptable.

Response 11: We thank the reviewer for their positive assessment of the Limitations section. We appreciate your feedback and have retained this section as presented.

Comment 12: Conclusions: I recommend developing a more critical analysis here, following the suggestions previously mentioned.

Response 12: We thank the reviewer for this valuable recommendation. In response, we have substantially revised the Conclusions section to include a more critical and reflective analysis, as suggested. The updated section now emphasizes the integration of clinical, economic, and psychosocial factors; highlights the importance of incorporating patient-reported outcome measures into routine practice and policy; and underscores the need for future longitudinal research and multidisciplinary approaches. We believe these enhancements better align the Conclusions with the expanded Introduction and Discussion and provide a stronger take-home message for clinicians, policymakers, and researchers.

Comment 13: References: Should be expanded according to the provided suggestions. I also recommend updating references that are more than 10 years old, unless they are methodological or of high impact in terms of evidence.

Response 13: We thank the reviewer for this helpful suggestion. In response, we have carefully reviewed and updated the reference list, replacing older citations with more recent and relevant literature published within the past 10 years, except for key methodological sources and high-impact foundational studies. We have also expanded the references to include additional recent studies related to chronic kidney disease management, dialysis care innovations, and patient-reported outcomes, as suggested. These updates have strengthened the scientific grounding and contemporary relevance of our manuscript.

Comment 14: In summary, the manuscript presents scientifically interesting results and, with the proposed improvements, could increase its overall quality. My recommendation is to proceed with a thorough revision, as, once properly modified, the manuscript could represent a valuable contribution to the relevant scientific literature.

Response 14: We sincerely thank the reviewer for this encouraging overall assessment and for recognizing the scientific value of our study. We greatly appreciate your thoughtful and constructive suggestions, which have helped us to substantially improve the clarity, depth, and overall quality of the manuscript. We have carefully implemented all recommended revisions and believe that the updated version represents a stronger and more valuable contribution to the literature.

Reviewer 4 Report

Comments and Suggestions for Authors

At the beginning, it should be noted that the text overlap is slightly more than 21%, which is a bit too much for an article!

- Replace with the term end-stage renal disease (ESRD)!

- The introduction is appropriate with satisfactory references!
- The methodology is adequately performed with Ethical approval and the date of
the decision and resolution.
The statistical analysis is correct!
- It is necessary to present a table with dialysis characteristics:
1) interdialysis yield
2) Kt/V
3) residual renal function
4) Virological status

- Add a table with the duration of chronic kidney disease and underlying renal disease!

- Are there and how many patients on the transplant list?

Author Response

We sincerely thank the reviewer for taking the time to review our manuscript and for providing thoughtful and constructive comments. We greatly appreciate your suggestions, which have helped us further improve the clarity and quality of our work.

Comment 1: At the beginning, it should be noted that the text overlap is slightly more than 21%, which is a bit too much for an article!

Response 1: We thank the reviewer for bringing this issue to our attention. Upon reviewing the overlap report, we would like to clarify that approximately 5% of the similarity originated from the Affiliations and Declarations sections, which contain standard institutional and funding information repeated across manuscripts submitted from the same research project. An additional 2% resulted from general background or methodological phrasing. The remaining 66 individual overlaps were each equal to or less than 1%, and did not represent substantive duplication.

Nonetheless, we carefully reviewed and revised the manuscript to further reduce overlap and improve originality. Specifically, we have substantially updated the Introduction, Results, and Discussion sections to enhance clarity, expand critical analysis, and reflect new content based on reviewer suggestions. We believe that the revised version now meets both scientific and ethical standards of originality.

Comment 2: Replace with the term end-stage renal disease (ESRD)!

Response 2: We thank the reviewer for this important suggestion to improve clarity and precision in terminology. In response, we have replaced general mentions of “kidney failure” with “end-stage renal disease (ESRD)” wherever applicable, to accurately reflect the patient population included in our study. Additionally, we have retained “chronic kidney disease (CKD)” when referring to earlier stages or to the disease continuum more broadly, in line with clinical practice guidelines. We believe these changes enhance the accuracy and scientific rigor of the manuscript.

Comment 3: The introduction is appropriate with satisfactory references!

Response 3: We thank the reviewer for this positive assessment of our Introduction. In addition to the original content, we have further expanded and updated the Introduction section based on the valuable suggestions from all reviewers, including adding new references to strengthen the clinical context and international perspective. We appreciate your feedback and believe these enhancements have improved the overall quality and relevance of the Introduction.

Comment 4: The methodology is adequately performed with Ethical approval and the date of the decision and resolution.

Response 4: We thank the reviewer for this positive evaluation of our methodological approach. We confirm that the study was conducted with appropriate ethical approval and that details regarding the approval number and date have been clearly presented in the manuscript. Additionally, we have further refined the Methods section to incorporate clarifications suggested by all reviewers during this revision process. We appreciate your feedback and support.

Comment 5: The statistical analysis is correct!

Response 5: We thank the reviewer for this positive assessment of our statistical analysis. We appreciate your feedback and are pleased that our analytical approach was found to be appropriate and rigorous.

Comment 6: It is necessary to present a table with dialysis characteristics: 1) interdialysis yield; 2) Kt/V; 3) residual renal function; 4) Virological status"

Response 6: We thank the reviewer for this valuable suggestion. Among the listed dialysis characteristics, we only assessed virological status (hepatitis B/C/D) in our study cohort. In response, we have added this information to Table 1 (Demographics and Clinical Characteristics) and have also re-analyzed our regression models in Table 5, including virological status as an additional predictor. We believe this addition strengthens the analysis and provides important clinical context.

Comment 7: Add a table with the duration of chronic kidney disease and underlying renal disease!

Response 7: We thank the reviewer for this helpful suggestion. In response, we have included detailed data on the duration of chronic kidney disease and underlying renal disease in Supplementary Tables 1-4 and Supplementary Figure 1. These additions provide a more comprehensive overview of the clinical characteristics of our study population and support the robustness of our analysis.

Comment 8: Are there and how many patients on the transplant list?

Response 8: We thank the reviewer for this insightful question. We confirm that we did not collect individual-level data on whether participants were actively on a transplant waiting list, as this was not part of our survey instrument. However, to provide context, we have added general national data in the Introduction: in Kazakhstan, there are approximately 11,000 patients on dialysis, among whom 3,748 are registered on the kidney transplant waiting list. We believe this addition offers important background on transplantation scarcity in the country.

Round 2

Reviewer 1 Report

Comments and Suggestions for Authors

The authors provide an improved revised version of the manuscript.

They better underlined the limits that the study has and the points of strenght.

I have no other considerations on the paper

Reviewer 3 Report

Comments and Suggestions for Authors

The authors have provided appropriate modifications to the manuscript.

Reviewer 4 Report

Comments and Suggestions for Authors

No